# Machine learning algorithms' application to predict childhood vaccination among children aged 12–23 months in Ethiopia: Evidence 2016 Ethiopian Demographic and Health Survey dataset

**Addisalem Workie Demsash**[1]*, **Alex Ayenew Chereka**[1], **Agmasie Damtew Walle**[1], **Sisay Yitayih Kassie**[1], **Firomsa Bekele**[2], **Teshome Bekana**[3]

**1** Department of Health Informatics, College of Health Science, Mettu University, Mettu, Ethiopia, **2** Department of Pharmacy, College of Health Science, Mettu University, Mettu, Ethiopia, **3** Biomedical Science Department, College of Health Science, Mettu University, Mettu, Ethiopia

* addisalemworkie599@gmail.com

**Data Availability Statement:** The dataset used for analysis is available on the DHS program website.

## Abstract

### Introduction

Childhood vaccination is a cost-effective public health intervention to reduce child mortality and morbidity. But, vaccination coverage remains low, and previous similar studies have not focused on machine learning algorithms to predict childhood vaccination. Therefore, knowledge extraction, association rule formulation, and discovering insights from hidden patterns in vaccination data are limited. Therefore, this study aimed to predict childhood vaccination among children aged 12–23 months using the best machine learning algorithm.

### Methods

A cross-sectional study design with a two-stage sampling technique was used. A total of 1617 samples of living children aged 12–23 months were used from the 2016 Ethiopian Demographic and Health Survey dataset. The data was pre-processed, and 70% and 30% of the observations were used for training, and evaluating the model, respectively. Eight machine learning algorithms were included for consideration of model building and comparison. All the included algorithms were evaluated using confusion matrix elements. The synthetic minority oversampling technique was used for imbalanced data management. Informational gain value was used to select important attributes to predict childhood vaccination. The If/ then logical association was used to generate rules based on relationships among attributes, and Weka version 3.8.6 software was used to perform all the prediction analyses.

### Results

PART was the first best machine learning algorithm to predict childhood vaccination with 95.53% accuracy. J48, multilayer perceptron, and random forest models were the

All the data generated and analyzed are included in the study.

**Funding:** The authors received no specific funding for this work.

**Competing interests:** The author declared that there are no competing interests in this work.

**Abbreviations:** ANC, Antenatal care; AUC, Area under the recursive curve; BCG, Bacillus Calmette Guerin; EDHS, Ethiopian Demographic and Health Survey; ROC, Receiver operators' curve; WHO, World Health Organization; DTP, Diphtheria Pertussis Tetanus.

consecutively best machine learning algorithms to predict childhood vaccination with 89.24%, 87.20%, and 82.37% accuracy, respectively. ANC visits, institutional delivery, health facility visits, higher education, and being rich were the top five attributes to predict childhood vaccination. A total of seven rules were generated that could jointly determine the magnitude of childhood vaccination. Of these, if wealth status = 3 (Rich), adequate ANC visits = 1 (yes), and residency = 2 (Urban), then the probability of childhood vaccination would be 86.73%.

## Conclusions

The PART, J48, multilayer perceptron, and random forest algorithms were important algorithms for predicting childhood vaccination. The findings would provide insight into childhood vaccination and serve as a framework for further studies. Strengthening mothers' ANC visits, institutional delivery, improving maternal education, and creating income opportunities for mothers could be important interventions to enhance childhood vaccination.

## Introduction

Globally, nearly 44% of child deaths occurred under 28 days of birth [1]. Around 75% of child deaths occur within 12 months of birth, and an estimated 4.1 million infants are projected to die in 2017 [2]. The rate of child deaths in developing countries is the highest in the world [2,3]. Around 1.2 million children are predicted to have died in Africa in the first 28 days of birth [4], and nearly 49% of child deaths are predicted to have occurred in Sub-Saharan countries [3]. According to the World Health Organization (WHO), more than half of child deaths are caused by infectious diseases that are easily preventable and treatable through simple and affordable interventions [5]. Worldwide, childhood mortality and morbidity are caused by tuberculosis, diphtheria, pertussis, tetanus, polio, and measles [6].

Child deaths due to diphtheria, pertussis, tetanus, polio, and measles are easily preventable through vaccines. Childhood vaccination is one of the most successful and cost-effective public health interventions for common childhood illnesses like pneumonia, diphtheria, tetanus, whooping cough, and measles [7]. Nowadays, nearly 3 million child deaths due to diphtheria, tetanus, whooping cough, and measles are prevented through child vaccination [8]. Over the past decade, more than 1,000,000 children's lives have been saved by immunization programs and infectious and communicable diseases have been controlled through child vaccination [9].

Nonetheless, 12.9 million children did not receive recommended vaccines across the world [8]. Sufficient numbers of children did not complete their immunization schedule due to various challenges and barriers [10]. Nearly 21 million children have been projected to miss out on vaccines, and two-thirds of vaccination missing occurred in developing regions due to the outbreaks of new cases [11]. Nearly 21 million children have been projected to miss out on vaccines, and two-thirds of the missing vaccinations occurred in developing regions due to an outbreak of new cases [12]. In Ethiopia, infant vaccination doses were usually delayed, with 63.8 percent of Diphtheria Pertussis Tetanus (DTP) dose 1, 63.1 percent of Polio dose 1, and 68.5 percent of measles delivered after the recommended date [13]. According to the Ethiopia Demographic and Health Survey (EDHS), data on vaccination coverage among children aged 12–23 months who received specific vaccines at any time before the survey revealed that only four out of ten children (43%) had received all basic vaccinations [14]. According to the WHO, the mean dropout rates of Bacillus Calmette–Guérin (BCG) and measles are 34.6% and 28.6%, respectively [15].

According to a traditional and multilevel logistic regression analysis report, different factors are reported that could affect child vaccination and immunization coverage. Maternal education, knowledge of mothers about the vaccines and their schedule, maternal age, fear of side effects, antenatal care (ANC) visits, and giving birth at a health institution are some of the maternally related factors that affect childhood vaccinations [16–19]. Additionally, the availability of vaccines, migration of caregivers, household income level, and sex of household heads are factors that affect childhood vaccination status [20,21]. Moreover, the sex and age of children, their birth interval and order, multiple children born at a time, a mother's media exposure, being a rural resident, and having distant health facilities are also factors associated with childhood vaccination [22,23]. However, the odds ratio and relative risk of traditional and multilevel logistic regression do not meaningfully classify attributes and do not discover new insights [24].

Despite the efforts of the government to improve child vaccination, increase vaccination coverage, and reduce vaccine dropout rates, vaccine providers and health programmers lack available on-site information handling tools to target high-risk children for vaccine dropout, and late and incomplete vaccination [15]. Therefore, low-income countries would model and visualize the childhood vaccination risks on large datasets to identify attributes for childhood vaccination and target children who are at a high risk of dropping out or delaying the next vaccine dose.

Massive amounts of biomedical and public health data are categorized and predicted using a variety of predictive algorithms to gain new knowledge and reveal hidden relationships and trends [25]. Multidimensional data mining techniques were used to correctly forecast future immunization outcomes based on existing data and to predict features of typical childhood immunization schedules [26]. Predictive analytics tools are potent and widely applicable for learning. Numerous machine learning algorithms have reportedly been used in earlier studies to predict disease prevalence, the use of healthcare services, vaccination uptake [27], routine immunization [15], childhood vaccination, and mortality [28,29]. For automated detection, identifying connections that aren't leaner, and identifying significant patterns in data, machine learning algorithms are essential [30].

Specifically, random forests, logistic regression, J48, logit boost, and Addaboost algorithms were used to predict under-five and neonatal mortality [29,31], undernutrition status of children [32], and malnutrition among children [33,34]. Additionally, Naïve Bayes and PART algorithms are also used to forecast and classify text documents [35]. Prediction of childhood vaccination based on machine learning techniques is insufficient. Currently, massive amounts of data are being generated. So, these must be presented with the best data analysis tools. Policymakers and stakeholders need accurate predictions on various aspects of immunization and other health parameters for effective actions. Researchers are needed to test and compare various prediction and classification algorithms that are needed to classify and predict childhood vaccination. Therefore, this study aimed to **(1)** evaluate different machine learning algorithms using model evaluation matrix parameters; **(2)** identify important attributes for childhood vaccination based on the best performance algorithm; and **(3)** generate association rules that predictor together determine the vaccination of children aged 12–23 months in Ethiopia.

## Methods and materials

### Study design and setting

A cross-sectional study design was conducted across the nine regions of Ethiopia. Ethiopia is located in the Horn of Africa and is bordered by Eritrea to the North, Djibouti and Somalia to the East, Sudan, and South Sudan to the West, and Kenya to the South. Ethiopia has nine

regional states with two administrative cities. These are subdivided into different administrative zones (817 Woredas and 16253 Kebeles) [36,37].

## Data source

The 2016 Ethiopian Demographic and Health Survey (EDHS) dataset was used from the DHS program website (https://dhsprogram.com). The survey was conducted by the Ethiopian Public Health Institute (EPHI) in collaboration with the Central Statistical Agency (CSA). The actual data collection period was conducted from January 18, 2016, to June 27, 2016.

## Sampling techniques and procedures

The sampling frame used for the 2016 EDHS is a frame of all Census Enumeration Areas (EAs) created for the 2016 Ethiopia Population and Housing Census (EPHC) and conducted by the Central Statistical Agency (CSA). The census frame is a complete list of the 84, 915 EAs, covering an average of 181 households, created for the 2016 EPHC. The sample for the 2016 EDHS was designed to provide estimates of key indicators for the country as a whole, for urban and rural areas separately, and for each of the nine regions and the two administrative cities. Two-stage stratified cluster sampling was used. Each region was stratified into urban and rural areas. In the selected EAs, a household listing operation was done, and the results were used as a sampling frame for household selection in the second stage. Finally, a fixed number of households per cluster was selected. Samples of EAs were selected independently in each stratum through implicit stratification and equal proportional allocation.

## Study populations

In this study, all living children aged 12–23 months were the source population, and all sampled living children aged 12–23 months living with their mothers were the study population. Details about the methodology of the data source, sampling procedure, and source population were presented in the 2016 EDHS report [38].

## Study variables

### Dependent variable

Childhood vaccination among children aged 12–23 months.

### Independent variables

Socio-demographic characteristics of households, such as wealth status, educational status of mothers, age of mother, region, residency, sex, and age of children, birth interval and birth order, sex of households' heads, ANC visit, place of delivery, working status, visiting health facility, and media exposure were used as independent attributes to predict childhood vaccination among children aged 12–23 months in Ethiopia.

## Operationalizations

### Childhood vaccination

Childhood vaccination among children aged 12–23 months was assessed using one dose of BCG, three doses of polio vaccine, three doses of DPT vaccine, and one dose of measles vaccine. Accordingly, the children had basic childhood vaccination if the children received at least one dose BCG vaccine, three doses of the polio vaccine, three doses of the DPT vaccine, and one dose of the measles vaccine, else children did not receive basic childhood vaccination.

Information on basic childhood vaccination status was obtained from (1) written vaccination record that includes infant immunization card and other health cards, (2) the mothers' verbal reports, and (3) health facility records [38].

### Birth interval

The period between two successive live births is a birth interval. For this study, a birth interval of <33 months between two consecutive live births is a **short birth interval**, whereas a birth interval of 33 and above is an **optimum birth interval** [39,40].

### ANC visits

The pregnant women had visited a health facility during their pregnancy for ANC services. Accordingly, the women had adequate ANC visits when the women visited the health facility at least four times for ANC services, otherwise inadequate ANC visits [41,42].

### Media exposure

If the mothers had access to either radio or television or both, then the mothers had media exposure; and if mothers did not any means of media access then the mothers had no media exposure.

### Data management and statically analysis

Data cleaning and labeling were performed using STATA version 15 software to prepare the data for analysis. Variables were recoded to meet the desired classification. To ensure the representativeness of survey results at the national level [43], sampling weights were applied during the analysis. The STATA version 15 software was used for data management and logistic regression analysis. Weka version 3.8.6 software was used for data pre-processing, important attribute selection that could predict childhood vaccination, and generating rules associated with childhood vaccination.

### Ethical approval and consent to participate

Ethical clearance was not necessary for this study since it was based on publicly available data sources. Informed consent from the study participants was also not applicable to this study. There are no attributes that uniquely identify individuals or households in this study. As a result, specific individuals, and households cannot be identified uniquely in this study according to the clinical study checklist (S1 File).

### Data pre-processing

Data pre-processing was used to manage missing and incomplete records, and duplicates. In the dataset, noise, outliers, and inconsistency are common. Therefore, all these unnecessary data values, including duplicate variables were managed. At this stage, all strings and categorical variables were transformed into nominal data types for ease of processing in Weka software.

### Feature selection

In this study, there were two stages of variable selection in the machine learning algorithm. In the first stage, a logistic regression analysis was employed for a feature or independent variables selection. A variable with a p-value of less than 0.2 with backward stepwise logistic

regression analysis was selected as a candidate for further important attribute selection. During the first phase of variable selection, a variance inflation factor was performed to check the correlation between variables. As a result, a variance inflation factor's value for all possible variables was less than four. Hence, there was no significant correlation between the variables. The Hosmer and Lemeshow tests were also performed to assess the model's fitness. Consequently, the model was fitted with a p-value of 0.263. In the second stage, a best-performance machine learning algorithm with information gain values was used to find important features or attributes that have a major contribution to predicting childhood vaccination among children aged 12–23 months in Ethiopia. The highest information gain value of an independent attribute is the most important attribute to predict childhood vaccination [44]. Then the next important attributes were selected based on their order of highest information gain value.

## Model building

### Data split and model selection

In this step of the machine learning algorithm, 70% of the datasets were used for training the model, and 30% of the datasets were used for testing the performance of the algorithms. A total of 1617 instances/ observations were included to predict childhood vaccination. From a total of 1617 observations, 1132 observations (70% of total observations) were used for training the model, and the remaining 485 observations (30% of total observations) were used for testing or evaluating the model. Various machine-learning algorithms were used to predict child mortality and health service utilization [25,33,34]. For this study, the various appropriate machine learning algorithms such as Naïve Bayes, PART, logistic regression, multilayer perceptron, J48, logit Boost, random forest, and AdaBoost were used to predict childhood vaccination among children aged 12–23 months in Ethiopia.

### Naïve Bayes

The Naïve Bayes algorithm is a supervised machine learning algorithm, which is based on the Bayes theorem and used for the classification and prediction of problems. In the Naïve Bayes algorithm, attributes are conditionally independent for the target class [25]. Naïve Bayes has a computational efficiency that several attributes and classification time is linear with several of several, and not affected by training time. Naive Bayes algorithms had an incremental learning behavior, could directly predict patterns with low variance, and their performance is measured by confusion matrix elements [45].

### PART

PART is a hybrid approach of a rule-based classification algorithm, and it uses a separate and conquer classification process [35]. It creates a partial decision tree from all the iterations and considers the suitable leaf into a rule. So, it is best to perform if/ then rules to extract and build knowledge for childhood vaccination [46].

### Logistic regression

Logistic regression is a type of regression model that is important to model the categorical dichotomous outcome variable or feature. Logistic regression is a statistical model used to classify and predict different parameters in health [47]. It might be a binary (Binary logistic) and (multiple) model used to predict binary (multiple) outcome variables. Logistic regression has different assumptions, of which the target variable is dichotomous, and independent variables that affect the target variable are indent of each other [48].

### J48 classifier algorithm

A J48 classifier algorithm is one of the best machine learning algorithms that examine categorical data based on a top-down recursive divide and conquer strategy [49]. J48 classifier is a simple C4.5 decision tree for classification to create a binary tree. The algorithm is crucial for classifying the problems, and the J48 algorithm is important to ignore the missing values and able to predict the item of missing value based on what is known about the records of another attribute. The process is to divide the available data into ranges based on the attribute values for that item that are found in the training data, and then classification is done and rules are generated from the attributes [50].

### Random forest

A random forest is a supervised machine-learning algorithm used to classify and problems health problems and health service utilization [51]. Random forest is the fastest to train and work with subsets of features, and it is important to detect complex relationships, including nonlinear and high-order interactions and yields the smallest prediction errors [52].

### Addaboost and logit boost

Addaboost is an ensemble meta-learning method that enhances the efficiency of the binary classification tree. Addaboost uses an iterative approach to learn from the mistakes of weak classifiers and turn them into strong ones [53,54]. AdaBoost is critical to boosting the performance of decision trees based on binary classification problems [55]. Another very powerful boosting classifier algorithm (**logit boost**) was used to predict childhood vaccination in this study. **The logit boost** algorithm is designed as an alternative solution to address the limitations of Addaboost in handling noise and outliers [56].

### Features of knowledge flow

The knowledge flow presents a "data-flow" inspired interface in Weka software for data processing and analysis. The knowledge flow can handle data either incrementally or in batches. The features of knowledge flow are initiative of the data flow layout, processing the data in batches or incrementally, processing multiple batches or streams in parallel, chain filtering together, and viewing and visualized model performance with a fold cross-validation [44]. The overall knowledge flow of model building for data processing, analyzing, and visualizing has been presented in **Fig 1**.

### Imbalance data management

Data imbalance mainly occurs in medical diagnosis, pattern recognition, speech, and fraud detection. The dataset might have majority and minority classes in its observation [57]. Therefore, the classification and prediction might be certain to the majority class. In such a case, the minority class might not be considered, and classification and prediction might be inaccurate and biased. Therefore, the synthetic minority over-sampling technique (SMOTE) was used to manage imbalanced data [58]. SMOTE creates new synthetic samples for the minority class by interpolating linearity between the minority class [58,59], and it is critical to address underfitting and overfitting to reduce prediction errors [60]. As a result, a total of 359 additional records were generated and added to the minority class. Overall, the imbalanced data and balanced data are presented in **Fig 2**.

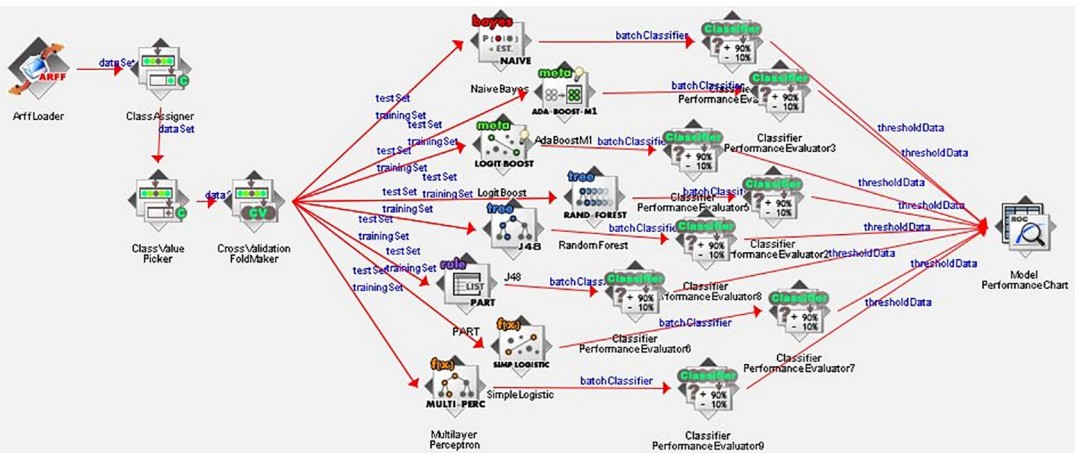

**Fig 1. Features of knowledge flow of the included algorithms.**

## Model evaluation

The performance of all the included algorithms has been evaluated using the confusion matrix. The accuracy of actual and predicted classes has been visualized by the confusion matrix model [61]. The predicted and actual classifications of under-five child mortality were compared using confusion matrix elements, such as true positive, false positive (FP), true negative, and false negative. The receiver operators' curve (ROC) was also used for model evaluation based on sensitivity, and specificity relationships. Since ROC is based on probability, the area under the ROC curve (AUC) is crucial to representing the degree or measure of separability. It tells how much the model is capable of distinguishing between classes. Hence, the higher the AUC, the better the model is at predicting true classes as true and false classes as false. Usually, the AUC value is good if it is greater than 80%, fair if it is between 70% and 80%, poor if it is between 60% and 70%, and failed if it is less than 60% [62]. A metric of interrater agreement i.e. kappa statistics was used to measure the degree of agreement/ reliability and to evaluate the accuracy of a classification. If the Kappa statistics value is ≤ 0 indicating the agreement is worse than random agreement, 0.01–0.20 slight agreement, 0.21–0.40 fair agreement, 0.41–0.60 moderate agreement, 0.61–0.80 substantial agreement, and 0.81–1.00 almost perfect agreement [63].

The formula for the confusion matrix's element is presented in **Box 1**.

---

Box 1. Formula for the element of the confusion matrix.

**Accuracy** $= (TP + TN)/(TP + TN + FP + FN)$

**Sensitivity** $= TP/(TP + FN)$

Not that Sensitivity = Recall = True Positive Rate (**TPR**)

**Specificity** $= TN/(TN + FP)$**False positive rate** $= FP/(FP + TN)$

**F_measure** $= 2TP/(2TP + FP + FN)$

**Precision** = **Postive predictive value** $= TP/2TP + FP)$

Whereas, **TP:** True positive, **TN:** True negative, **FP:** False positive, **FN:** False negative

---

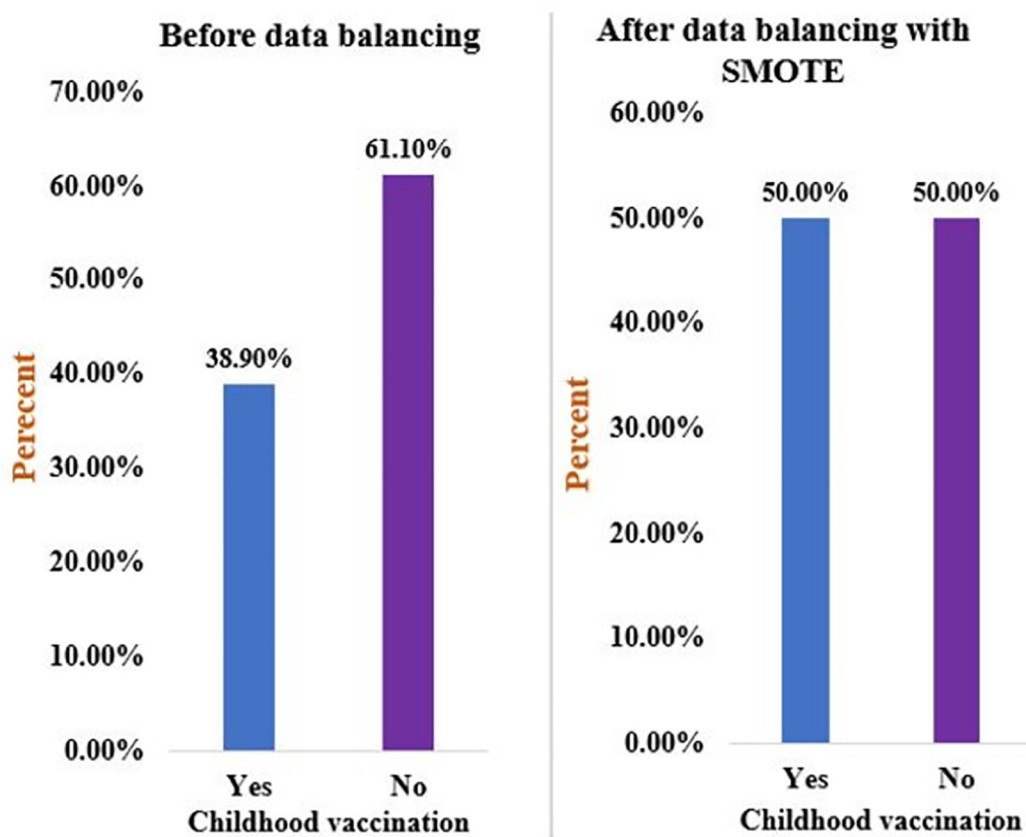

**Fig 2. Overall childhood vaccination status among children aged 12–23 months in Ethiopia, before and after data balancing, using the 2019 EDHS dataset.**

**True positive:** The model correctly predicts a positive class of response outcome.

**False positive:** The model incorrectly predicts a positive class in the response outcome.

**True negative:** The model correctly predicts a negative class in the response outcome.

**False-negative:** The model incorrectly predicts a negative class in the response outcome.

**Sensitivity:** Sensitivity is the test to measure correctly positive predicted events out of a total number of positive events, and it shows the value of how many positives are predicted out of total positive classes.

**Specificity:** Specificity is the proportion of real negative cases that were predicted as negative. This indicates that there will be another proportion of real negative cases, which would be predicted as positive and could be termed as false positives.

**Precision:** Precision is a positive predictive value, and it is the correct events divided by the total number of positive events that the classifier predicts.

**F_measure:** F measure is the inverse relationship between accuracy and recall. The higher value of the F-measure score predicts a better model.

## Prediction and association rule mining

Once the model is built and its performance assessed, childhood vaccination among children aged 12–23 months is predicted based on the predictors. Important variables selected based on a best-performance model were used to predict childhood vaccination. Although important

variables are used to predict childhood vaccination, the predictive model does not show which nominal variables are jointly associated with childhood vaccination among children aged 12–23 months.

Therefore, association rule mining analysis (the **If** (antecedent)/ **then** (consequent) statements) is used to discover relationships between seemingly relational attributes. Association rule mining analysis is important for non-numerical and categorical types of data attributes. It is important to observe frequently occurring patterns and identify the dependencies between attributes by supporting how frequently the if/then relationship appears in the observations and confidence in the number of times the relationships are true. The if/ then association rule mining analysis is critically important to select important features that jointly determine childhood vaccination and is the easiest way to interpret [64].

For the association rule mining analysis, the apriori algorithm method was used to identify strong and frequently related attributes. The **If then** association rule is the pair of X and Y (X, Y) attributes expressed as X->Y, where X is an antecedent and Y a consequent that is as X happens Y would also happen [65]. These rules are critically important for the prevention and control of health problems and crucial for health policymakers' proactive decision-making purposes. Various studies have widely used if/then rules in healthcare research, such as predicting childhood care and child mortality [66], predicting parasite infection [67], the pattern of new cases and stroke [68,69], and maternal healthcare service utilization discontinuation to identify important features [70]. The relationship between X and Y attributes is expressed in the following way [69].

If the left attribute $>1$|X and Y are positively associated to determine childhood vaccination. if the left attribute $<1$|X and Y negatively associated to determine childhood vaccination.

If the left attribute = $1$|No relation between X and Y to determine childhood vaccination.

The detail of data preparation, model building, important variable selection, and analysis workflow is presented in **Fig 3**.

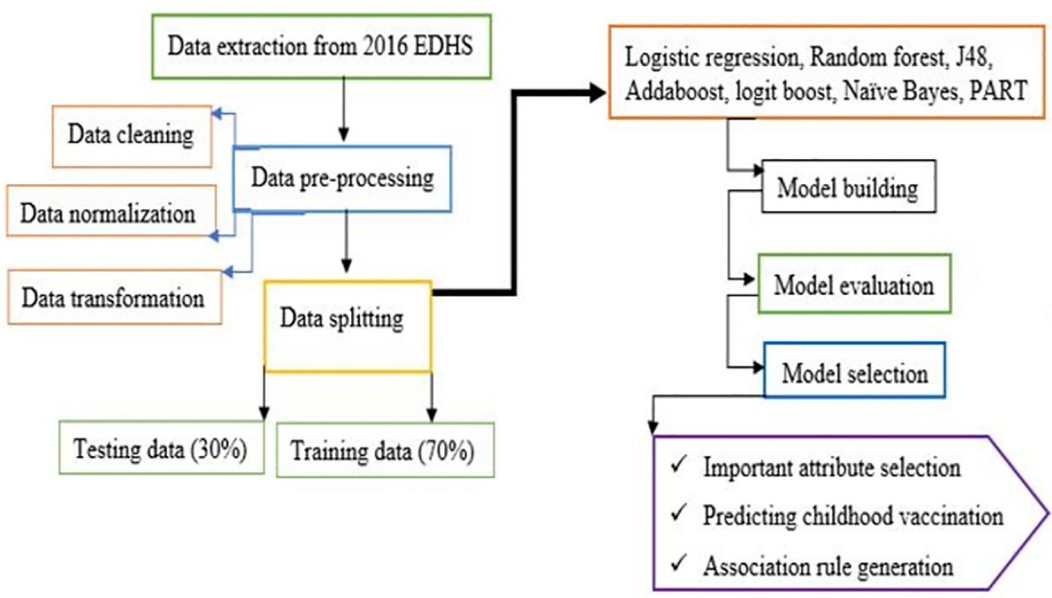

**Fig 3. Workflow for data pre-processing, and childhood vaccination prediction processing.**

# Results

## Children's and mothers' characteristics

A total of 1617 weighted samples of children aged from 12–23 months were included for analysis. The majority (62.52%) of children's mothers were under the age of 35 years. The majority (72.5%) of children were born from mothers who had not had formal education. Seven hundred thirty (45.1%) and two hundred eighty-eight (17.8%) children were from the Oromia and Amhara regions, respectively. The majority (91.2%) of the children were born to rural residents' mothers. Five hundred fifty-three (34.2%) and four out of ten (40.3%) of children were born from mothers whose religious were Orthodox and Muslim, respectively. Seven hundred sixty (47%) of children's mothers were poor. Nearest to half (52.9%) and the majority (86.2%) of children and household heads were female and male, respectively. Six hundred seventy-five (41.7%) of children were under the age of 12–15 months (**Table 1**).

## Children's and mothers' characteristics

Less than half (47.2%) of children visited a health facility in the last 12 months after birth, and the majority (70.6%) of children's mothers had not worked during the time of the interview. Only 29.8% of children's mothers had media exposure, and 29% of mothers had given birth to health institutions. The majority (70.6%) of the mothers did not adequate ANC visits during their pregnancy period. The majority (64.5%) of children had a birth order of less than five, and 65.1% of children had an optimal birth interval (**Fig 4**).

## Vaccination coverage among children aged 12–23 months in Ethiopia

In Ethiopia, the overall vaccination coverage of children aged 12–23 months was 38.9% (95% CI: 36.52%-41.28%). Specifically, more than half (54.1%) of children received the measles vaccination, and seven out of ten (68.6%) of children had received the BCG vaccine. The majority (72.9%), nearly two-thirds (65.2%), and more than half (53%) of children had received DPT1, DPT2, and DPT3 vaccines, respectively. The majority (80.8%), nearly seven out of ten (72%), and more than half (55.9%) of children aged 12–23 months had received POLIO 1, POLIO 2, and POLIO 3 respectively (**Fig 5**).

## Models performance to predict childhood vaccination in Ethiopia using 2016 EDHS data

Eight machine learning algorithms were used to predict childhood vaccination in Ethiopia. The PART, Naïve Bayes, logit boost, J48, random forest, addaboost, logistic regression, and multilayer perceptron algorithms were included to predict childhood vaccination. The confusion matrix parameter elements (TPR, FNR, precision, F-measure, AUR, and accuracy) were used to evaluate the performance of the included algorithms. Accordingly, the PART algorithm was the first best performance algorithm to predict childhood vaccination with 95.53% accuracy, and 91.89% of AUC. The Kappa statistics value also confirmed that the classification accuracy of the PART algorithm was almost perfect with 86.57% of accuracy. The j48 algorithm was the second-best machine learning algorithm to predict childhood vaccination with 89.24% accuracy. The 86.01% AUR value also confirmed that the j48 algorithm was the best model next to the PART algorithm, and the classification accuracy of the j48 algorithm had a substantial agreement with 79.27% of kappa statistics. The overall machine learning algorithms comparison for childhood vaccination are presented in **Table 2** and **Fig 6**.

**Table 1. Children's and mothers' characteristics, 2016 EDHS data (n = 1617).**

| Variable | Category | Frequency (n) | Percent (%) |
|---|---|---|---|
| Mothers' educational status | No formal education | 1172 | 72.5 |
| | Primary | 377 | 23.3 |
| | Secondary | 39 | 2.4 |
| | Higher | 29 | 1.8 |
| Region | Tigray | 116 | 7.2 |
| | Afar | 16 | 1.0 |
| | Amhara | 288 | 17.8 |
| | Oromia | 729 | 45.1 |
| | Somali | 60 | 3.7 |
| | Benishangul | 18 | 1.1 |
| | SNNPR | 347 | 21.4 |
| | Gambela | 4 | .2 |
| | Harari | 3 | .2 |
| | Addis Ababa | 29 | 1.8 |
| | Dire Dawa | 7 | .4 |
| Mothers' age (year) | 15–34 | 1011 | 62.52 |
| | > = 35 | 606 | 37.48 |
| Family's wealth index | Poor | 760 | 47.0 |
| | Middle | 339 | 20.9 |
| | Rich | 518 | 32.1 |
| Mother/caregiver religion | Orthodox | 553 | 34.2 |
| | Catholic | 26 | 1.6 |
| | Protestant | 346 | 21.4 |
| | Muslim | 652 | 40.3 |
| | Traditional, and other | 40 | 2.5 |
| Place of residency | Urban | 142 | 8.8 |
| | Rural | 1475 | 91.2 |
| Sex of children | Male | 761 | 47.1 |
| | Female | 856 | 52.9 |
| Sex of household head | Male | 1394 | 86.2 |
| | Female | 223 | 13.8 |
| The current age of children (months) | 12–15 months | 675 | 41.7 |
| | 16–19 months | 519 | 32.1 |
| | 20–23 months | 423 | 26.2 |

## Importance attributes of childhood vaccination in Ethiopia

The information gain coefficients with a 10-cross-fold validation process were used to select important attributes of childhood vaccination in Ethiopia. The best performance model (PART algorithm) was used to select important attributes for childhood vaccination. According to the PART algorithm report, having adequate ANC visits, institutional delivery, visiting health facilities in the last 12 months, higher educational status of mothers, children whose mothers were rich, being of urban residents, female household heads, mothers' age greater than 35 years, having birth order less than five, and mothers currently working were important attributes for childhood vaccination among children aged 12–23 months. The important attributes and their information gain values are presented in **Table 3** and **Fig 7**.

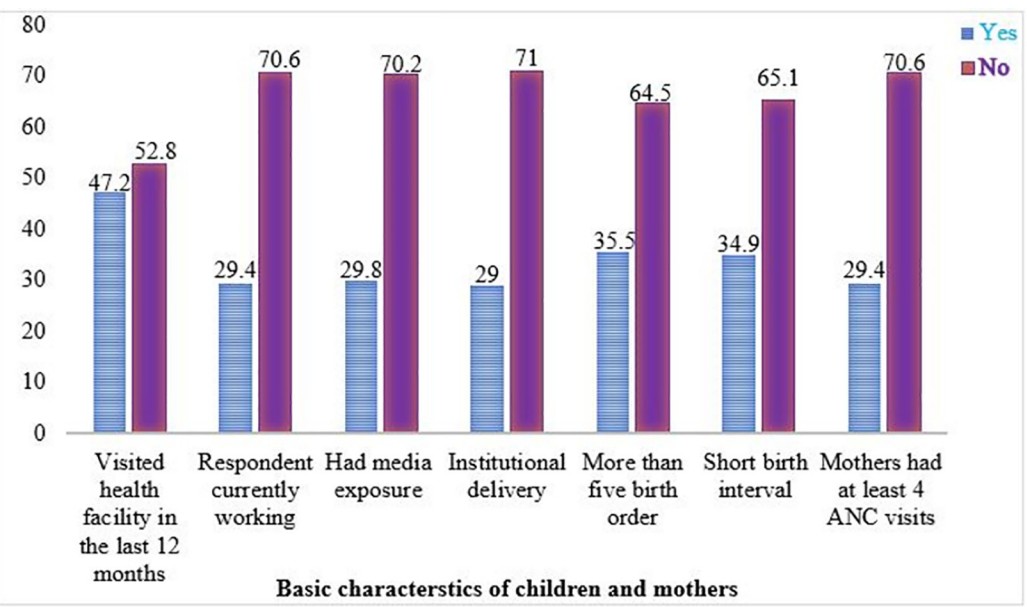

**Fig 4. Children's and mothers' characteristics.**

## Association rule building

The association rule generation process was done based on important attributes selected by performing the best-performing machine learning model (PART). A total of seven association rules were generated, and the details of the rules were presented in Box 2.

---

### Box 2. Association rule generation and knowledge extraction

**Rule 1: If** wealth status = 3 (Rich), adequate ANC visits = 1 (Yes), and residency = 2 (Urban), **then** the probability of childhood vaccination would be 86.73% (left = 1.87).

**Rule 2: If** institutional delivery = 1 (Yes), mothers' educational status = 4 (Higher), and household heads' sex = 0 (Female), **then** the probability of childhood vaccination would be 82.14% (left = 1.67).

**Rule 3: If** adequate ANC visit = 1 (Yes), mothers' age = 1 (>35 years), and institutional delivery = 1 (Yes), **then** the probability of childhood vaccination would be 79.21% (left = 1.47).

**Rule 4: If** birth order >5 = 0 (No), visited HF in the last 12 months = 1 (Yes), residency = 2 (Urban) and mothers' current working = 1 (Yes), **then** the probability of childhood vaccination would be 66.81% (left = 1.32).

**Rule 5: If** institutional delivery = 1 (Yes), mothers' wealth status = 2(Middle), and visited HF in the last 12 months = 1 (Yes), **then** the probability of childhood vaccination would be 62.45% (left = 1.25).

**Rule 6: If** residency = 2 (Urban), birth order >5 = 1 (Yes), wealth status = 2 (Middle), and adequate ANC visits = 1 (Yes), **then** the probability of childhood vaccination would be 57.16% (left = 1.17).

**Rule 7: If** mothers' educational status = 3 (Secondary), institutional delivery = 1 (Yes), and mothers currently working = 1 (Yes), **then** the probability of childhood vaccination would be 51.92% (left = .12).

---

**Table 2. Model accuracy of the included machine learning algorithms Based on confusion matrix parameters.**

| Confusion matrix Parameters (%) | The included machine-learning algorithms | | | | | | | |
|---|---|---|---|---|---|---|---|---|
| | PART | Naïve Bayes | Random forest | Logit Boost | J48 | AdaBoost | Multilayer perceptron | LR |
| True positive rate (%) | 89.90 | 64.00 | 88.50 | 72.10 | 88.40 | 69.80 | 83.60 | 70.90 |
| False positive rate (%) | 18.20 | 30.80 | 1.50 | 38.90 | 32.40 | 35.21 | 12.10 | 36.30 |
| Precision (%) | 93.80 | 73.90 | 87.80 | 70.60 | 76.00 | 72.00 | 81.00 | 71.70 |
| F-measure (%) | 94.30 | 68.20 | 88.60 | 71.30 | 77.71 | 70.90 | 82.30 | 71.30 |
| Relative absolute error (%) | 51.78 | 75.33 | 34.05 | 83.68 | 75.05 | 85.10 | 23.19 | 84.21 |
| AUC (%) | 91.89 | 72.30 | 82.70 | 73.20 | 86.01 | 72.90 | 83.29 | 65.60 |
| Kappa statistics (%) | 86.57 | 32.66 | 78.68 | 33.26 | 79.27 | 36.50 | 72.02 | 35.00 |
| Accuracy (%) | 95.53 | 66.29 | 82.37 | 77.28 | 89.24 | 68.60 | 87.20 | 65.80 |
| Note that LR, stands for Logistic Regression | | | | | | | | |

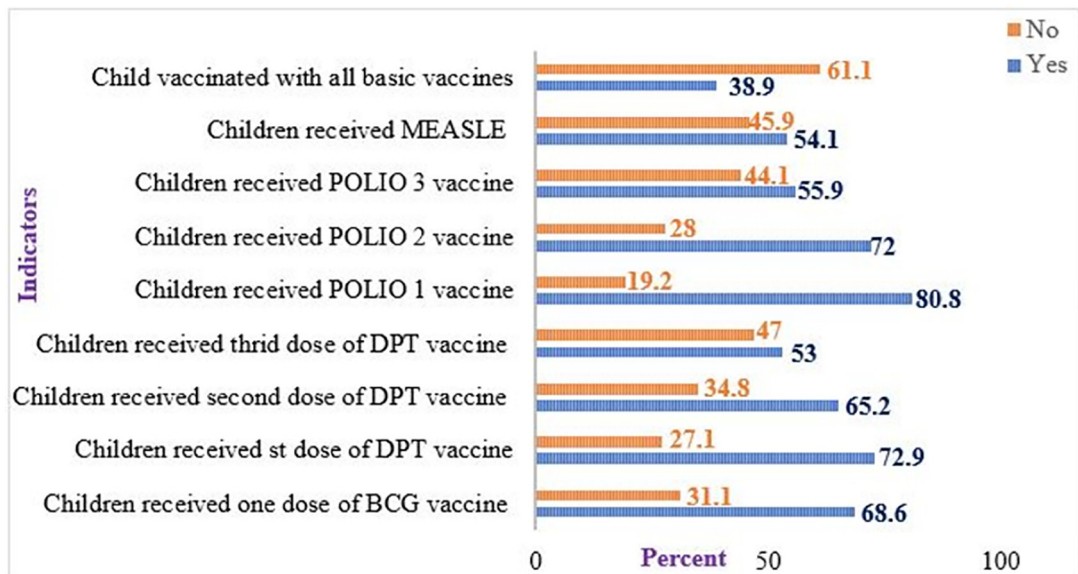

**Fig 5. The vaccination status of children aged 12–23 months with recommended vaccination types.**

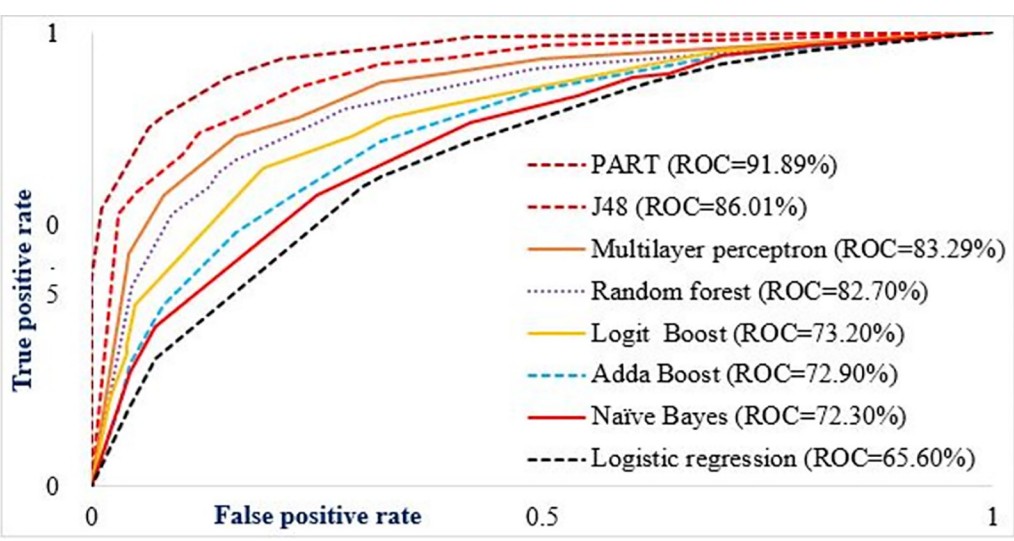

**Fig 6. Comparison of machine learning algorithms using the area under ROC value.**

**Table 3. Information gain value for each predictor.**

| Predictor variables | Type | Measurement | Information gain value |
|---|---|---|---|
| Adequate ANC visit, (Yes) | Nominal | Scale | 0.087 |
| Institutional delivery, (Yes) | Nominal | Scale | 0.084 |
| Visited HF in the last 12 months, (Yes) | Nominal | Scale | 0.076 |
| Educational status, (Higher) | Nominal | Scale | 0071 |
| Wealth status, (Rich) | Nominal | Scale | 0.071 |
| Residency, (Urban) | Nominal | Scale | 0.062 |
| Household sex, (Female) | Nominal | Scale | 0.059 |
| Mothers' age, (>35 years) | Nominal | Scale | 0.037 |
| Birth order >5, (No) | Nominal | Scale | 0.023 |
| Mothers currently working, (Yes) | Nominal | Scale | 0.023 |

## Discussion

The 2016 EDHS dataset was used, with a total of 1617 sampled observations. The childhood vaccination status of children aged 12–23 months was assessed. As a result, nearly four out of ten (38.9%) of children had received at least one dose of the BCG vaccine, three doses of the polio vaccine, three doses of the DPT vaccine, and one dose of the measles vaccine. The current finding was higher than the study done in the Dabat demographic and health survey site, in Ethiopia [71]. According to the World Health Organization vaccination estimation, the current finding was inadequate since below 90%. Plus, the finding was lower than the study done in East Africa, 69% [72], and in Gondar City, 98% [73]. This might be due to disparities in vaccination program access, and mothers might not understand the value of childhood vaccinations, and not remember when the children had been appointed [74]. Additionally, different natural and human-made factors might limit the uptake of childhood vaccination [75,76]. Moreover, women might face problems with health service access (70.2%), might have poor health-seeking behavior, high transportation costs, and inaccessibility of health facilities might be significant reasons for low coverage of childhood vaccination in Ethiopia [77].

70% and 30% of total observations were set for model training, and model evaluation, respectively. The objectives were to evaluate machine learning algorithms and to identify the

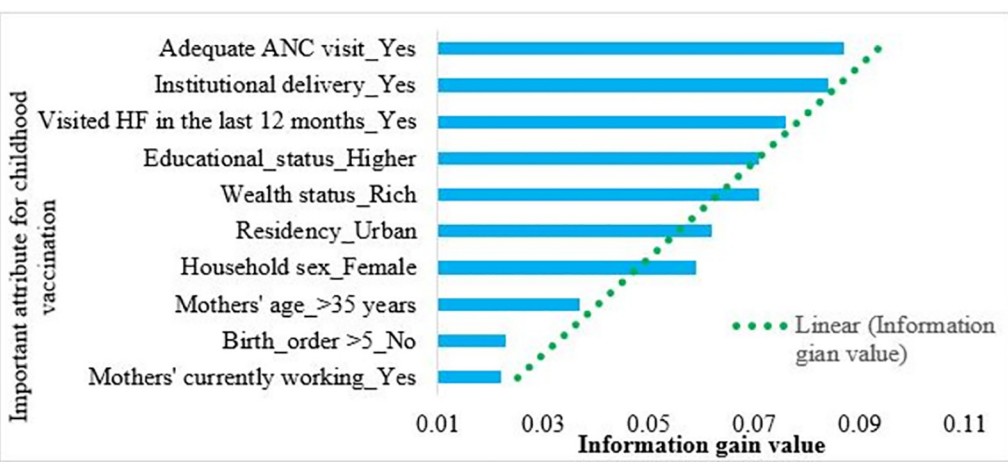

**Fig 7. Important attributes selection, based on best performance algorithm (PART), to predict childhood vaccination among children aged 12–23 months in Ethiopia.**

best algorithm to select important attributes to predict childhood vaccination in Ethiopia. Hence, eight machine learning algorithms were considered for comparison. Different confusion matrix elements were used to compare the candidate machine learning algorithms.

The included eight machine learning algorithms were evaluated and compared by classification matrix elements accuracy and AUR score values. Hence, the accuracy and AUR value of the PART algorithms were 95.53% and 91.89% with 10-fold cross-validations, respectively. Hence, the PAR algorithm was the first accurate model to predict childhood vaccination among children aged 12–23 months in this study. This finding was agreed with studies done about data classification and terms of association [35], and the application of data mining for the prediction of patients' CD4 count [78]. The j48, multilayer perceptron, and random forest algorithms were the second, third, and fourth best machine learning algorithms to predict childhood vaccination with 89.24%, 87.20%, and 82.37% accuracy, respectively. This finding was supported by various studies conducted to predict under-five child mortality [29,32,44,79], contraceptive discontinuation [70], stunting, and malnutrition among children [80–82].

The second objective of the study was to select important attributes that could predict childhood vaccination among children aged 12–23 months in Ethiopia. From the attributes selected to predict childhood vaccination, adequate ANC visits, institutional delivery, health facility visits, higher education of mothers, rich wealth status, children from urban areas, female household heads, a mother's age greater than 35 years, a child's birth order less than five, and mothers currently working were important attributes to predict vaccination of children aged 12–23 months in Ethiopia.

Adequate ANC visits were the top-ranked attribute to predict childhood vaccination among children aged 12–23 months in Ethiopia, with a 0.087 information gain value. This finding was agreed upon with the previous similar studies done in Ethiopia [14,83], and Zimbabwe [84]. This might be due to women who attend ANC follow-up might get counseling services about child immunization [85], and mothers might receive adequate education about the importance of postnatal visits [86]. Moreover, an adequate number of ANC visits is associated with a greater likelihood of having a child vaccinated [87].

Institutional delivery was the second-most important attribute in predicting childhood vaccination. This finding is supported by similar studies done in Ethiopia [8,14], and Nigeria [88]. This might be because children who were born at health facilities might be more likely to get BCG and OPV 0 vaccines at birth than children who were born elsewhere [85]. Plus, institutional delivery might create an opportunity for children's mothers to communicate with health professionals about the importance and side effects of immunization, and the vaccine initiation time [85]. Moreover, children's mothers who gave birth at health facilities might get information about the basic childhood vaccination services for the current and the next vaccination appointment schedules [89].

Visiting HF was the third most important attribute in predicting childhood vaccination in Ethiopia. This finding was in line with studies done in Ethiopia [14], and similar resource-limited settings [90,91]. This might be because mothers who visit a health facility might receive adequate education and counseling about child immunization, and mothers after birth are recommended to visit a health facility for postnatal check-ups and services [85].

The higher educational status of mothers was the fourth important attribute to predict childhood vaccination among children aged 12–23 months. This study is also similar to a study done in Bangladesh in that maternal education is an important feature in predicting anemia among under-five children [92]. Another study done in India also supports the current findings of the study [93]. This might be due to educated mothers knowing the importance of

vaccines for child care, and educated mothers empowering them and feel free to make decisions to visit the health facility for child health services [94].

Being rich and being urban residents were the fifth and sixth important attributes to predict childhood vaccination among children aged 12–23 months in Ethiopia. This finding was similar to a study done in Bangladesh [92], and Ethiopia [8]. This might be because mothers from urban areas might have more access to media, which plays a vital role in disseminating educational information and creating awareness [95,96]. Plus, children's mothers in an urban area might have adequate information communication technology infrastructure that enables them to receive short message services for health information services access [1]. Therefore, children in urban areas might be more likely to get and uptake vaccines. Moreover, wealthier people might have media access and afford to cover the transport cost of health facilities, so they might have access to information and better health-seeking behavior, and good childcare practices [71].

Generating rules for childhood vaccination was the third objective of the study. Previous studies have assessed the joint effect of independent predictors on the outcome of interest [44,70,78]. Consequently, seven association rules were generated to determine vaccination status among children aged 12–23 months in Ethiopia. According to association rule 1, the probability of a childhood vaccination would be 86.73%, if and only if the mothers' wealth status was rich, mothers had adequate ANC visits, and the children were urban residents. This might be because women with rich wealth status might be able to afford to pay any costs needed for vaccination, mothers who had adequate ANC visits might have adequate awareness and knowledge about child vaccination during their health facility visits during their pregnancy period, and health facilities in urban areas might be easily accessible for mothers to vaccinate their children. The effects of these three attributes are critical for childhood vaccination, and the combination of these factors might make it particularly important for children to be vaccinated when they are under 12 to 23 months. Based on Rule 2, childhood vaccination would be 82.14%, if mothers gave birth at health institutions, mothers' educational status was higher, and if the household heads' sex was female. The if/ then rules are critical to discovering hidden relationships between attributes, extracting knowledge from a set of data, and accurately representing knowledge and information about the vaccination of children. The findings presented in this study are critically important for policymakers and stakeholders to support public health action, decision-making purposes, and the storage of knowledge regarding child vaccination status.

## Strengths and limitations of the study

In this study, machine learning algorithms are used to classify, and predict childhood vaccination. This study used nationally representative data, and the findings might be representative of the study populations. However, machine learning algorithms do not have coefficients like odds and incident rate ratios. Therefore, the strength and direction of associations are unknown.

## Conclusions

In this study, PART, J48, multilayer perceptron, and random forest algorithms were the first, second, third, and fourth best performance machine learning algorithms to predict childhood vaccination in Ethiopia. Adequate ANC visits, institutional delivery, health facility visits, higher educational status, and rich mothers were the top five important attributes to predict childhood vaccination in Ethiopia. Moreover, seven rules were generated that attributes together can determine the magnitude of childhood vaccination.

The findings of this study would support policymakers and stakeholders in developing childcare intervention mechanisms and early preparedness for caring for children through child immunization, and the findings would serve as input for immunization coverage and reduction of vaccine dropouts. The generated rule would be important for knowledge creation and representation. Specifically, stakeholders are recommended to enhance mothers' ANC visits and institutional delivery by constructing nearby health facilities. Creating income opportunities and awareness of mothers would be also critical interventions for childhood vaccination. Moreover, the current study would serve as a baseline for future studies.

## Supporting information

**S1 File. Checklist for the current study.**
(PDF)

## Acknowledgments

The authors would like to express their deepest appreciation to the DHS program for permitting data access and use for this study.

## Author Contributions

**Conceptualization:** Addisalem Workie Demsash, Teshome Bekana.

**Data curation:** Addisalem Workie Demsash, Teshome Bekana.

**Formal analysis:** Addisalem Workie Demsash, Teshome Bekana.

**Investigation:** Addisalem Workie Demsash.

**Methodology:** Addisalem Workie Demsash, Teshome Bekana.

**Validation:** Addisalem Workie Demsash.

**Writing – original draft:** Addisalem Workie Demsash, Alex Ayenew Chereka, Agmasie Damtew Walle, Sisay Yitayih Kassie, Firomsa Bekele.

**Writing – review & editing:** Addisalem Workie Demsash, Alex Ayenew Chereka, Agmasie Damtew Walle, Sisay Yitayih Kassie, Firomsa Bekele.

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
