## [Decision Letter · Decision Letter 0]

19 Jun 2023

PONE-D-23-09622Predicting childhood vaccination among children aged 12-23 months in Ethiopia: Using machine learning algorithmsPLOS ONE

Dear Dr. Addisalem Workie Demsash

Thank you for submitting your manuscript to PLOS ONE. After careful consideration, we feel that it has merit but does not fully meet PLOS ONE’s publication criteria as it currently stands. Therefore, we invite you to submit a revised version of the manuscript that addresses the points raised during the review process.

We look forward to receiving your revised manuscript.

Kind regards,

Engidaw Fentahun Enyew, MSc

Academic Editor

PLOS ONE

Journal Requirements:

 Whilst you may use any professional scientific editing service of your choice, PLOS has partnered with both American Journal Experts (AJE) and Editage to provide discounted services to PLOS authors. Both organizations have experience helping authors meet PLOS guidelines and can provide language editing, translation, manuscript formatting, and figure formatting to ensure your manuscript meets our submission guidelines. To take advantage of our partnership with AJE, visit the AJE website (http://aje.com/go/plos) for a 15% discount off AJE services. To take advantage of our partnership with Editage, visit the Editage website (www.editage.com) and enter referral code PLOSEDIT for a 15% discount off Editage services. If the PLOS editorial team finds any language issues in text that either AJE or Editage has edited, the service provider will re-edit the text for free.

 A clean copy of the edited manuscript (uploaded as the new *manuscript* file).

“Financial supports were not recieved fro this study”

Additional Editor Comments (if provided):

please, address all reviewer comments

Reviewers' comments:

Reviewer's Responses to Questions

**Comments to the Author**

1. Is the manuscript technically sound, and do the data support the conclusions?

Reviewer #1: Yes

Reviewer #2: Yes

2. Has the statistical analysis been performed appropriately and rigorously? 

Reviewer #1: Yes

Reviewer #2: Yes

3. Have the authors made all data underlying the findings in their manuscript fully available?

Reviewer #1: Yes

Reviewer #2: Yes

4. Is the manuscript presented in an intelligible fashion and written in standard English?

Reviewer #1: Yes

Reviewer #2: Yes

5. Review Comments to the Author

Reviewer #1: This type of methodological modeling research in operational research is not as such common and I appreciate the authors to come up with such insightful new approach.

Few concerns and suggestion for the authors: Q1? It was better if you make the presentation those statistical modelling easier for all readers to read.

Q2? Through this machine learning algorithms approach of identifying predictors of child vaccination, what are new from the usual operatonal research approach

Reviewer #2: Review Reports

Title: Predicting childhood vaccination among children aged 12-23 months in Ethiopia: Using machine learning algorithms

Manuscript ID: PONE-D-23-09622

Review Comments

Are you predicting vaccination uptake or whether the children have taken vaccinations? Or the outcome of the vaccinations? Why do we predict? Is there no other means to gain this data?

The abstract section needs major revision E.g., Mixed reporting in the methods and result section and the key terms are incomplete

Which model was used? It is inconsistent in the methods and results section.

Shorten and add efforts made for improving vaccination in Ethiopia?

The study area is NOT referenced.

In the operational definitions try to be specific E.g., is that access to media or use of the medias?

In the model building section appropriate references are lacking

You can rewrite “Children’s and mothers’ characteristics” for example as “Socio-demographic Characteristics of Children’s and mothers” and avoid inconsistency e.g., basic characteristics….

The tables and figures are not self-explanatory. In addition, the figure is not referenced

Try to see this sentence again “This might be because some vaccines such as, BCG and OPV 0 are often 460 given immediately after birth at health facilities [70].” With the percentage of institutional delivery. Similarly, try to revisit “household head 491 was female” because it is very law and is associated with your statistical efficiency.

The result section needs brief revision.

The discussion section is inadequately discussed and reasoned out.

The recommendations and the conclusions should be context based and practical.

For your research carrier try to do with others/team which is one component of professionalism

Regards,

6. PLOS authors have the option to publish the peer review history of their article (what does this mean?). If published, this will include your full peer review and any attached files.

Reviewer #1: **Yes: **Abebe Sorsa

Reviewer #2: No

---

## [Author Response · Author response to Decision Letter 0]

30 Jun 2023

Dear editor and reviewer, I have uploaded the reviewers' response. Please find the detial response in the uploaded file.

---

## [Editor Report · Decision Letter 1]

6 Jul 2023

Machine learning algorithms’ Application to predict childhood vaccination among children aged 12-23 Months in Ethiopia: Evidence 2016 Ethiopian Demographic and Health Survey Dataset

PONE-D-23-09622R1

Dear Dr. Demsash Addisalem Workie 

We’re pleased to inform you that your manuscript has been judged scientifically suitable for publication and will be formally accepted for publication once it meets all outstanding technical requirements.

Kind regards,

Engidaw Fentahun Enyew, MSc

Academic Editor

PLOS ONE
---

## [Editor Report · Acceptance letter]

5 Oct 2023

PONE-D-23-09622R1 

Machine learning algorithms’ Application to predict childhood vaccination among children aged 12-23 Months in Ethiopia: Evidence 2016 Ethiopian Demographic and Health Survey Dataset 

Dear Dr. Demsash:

I'm pleased to inform you that your manuscript has been deemed suitable for publication in PLOS ONE. Congratulations! Your manuscript is now with our production department. 

Kind regards, 

on behalf of

Engidaw Fentahun Enyew 

Academic Editor

PLOS ONE